*The Company of*
**Biologists**

## RESEARCH ARTICLE

# Kindlins regulate integrin- and growth factor-dependent ureteric bud formation

Shensen Li[1,*], Fabian Bock[1,2,3,*], Olga Viquez[1], Anjana Hassan[1], Sijo Mathew[1,4], Riya Palamuttam[1,4], Glenda Mernaugh[1], Xinyu Dong[1,2], Meiling Melzer[1], Matthew Tantengco[1], Thomas Carroll[5], Andrew Terker[1], Juan Pablo Arroyo[1], Ambra Pozzi[1,3,6] and Roy Zent[1,2,3,‡]

## ABSTRACT

The kidney collecting system develops from the ureteric bud (UB), which undergoes multiple rounds of iterative branching. This process is controlled by growth factors and requires the interaction between the extracellular matrix and β1-containing integrin receptors. Integrin affinity for its ligands is regulated by integrin-binding proteins including kindlins, which bind well-defined motifs within the β subunit cytoplasmic tail. We show that mice expressing β1 integrins with mutations that abrogate kindlin binding in the developing UB have mild medullary hypoplasia and a moderate branching defect. Collecting duct (CD) cells expressing the same mutations in the β1 subunit have moderate tubulogenesis, spreading and adhesion defects, but show intact growth factor-dependent signaling. In contrast, mice lacking kindlins in the UB are anephric due to a complete absence of UB budding. Kindlin-knockout CD cells are unable to spread, adhere or respond to growth factors, irrespective of whether the integrins are bound to a ligand. Thus, in addition to regulating integrin function, kindlins mediate crucial growth factor signaling required for initial UB formation.

KEY WORDS: Kidney development, Branching morphogenesis, Integrins, Kindlins, Growth factor

## INTRODUCTION

The kidney collecting system develops from the ureteric bud (UB), which undergoes multiple rounds of iterative branching in a process called branching morphogenesis. In mice, this is initiated at embryonic day 10.5, when the Wolffian duct grows into the metanephric mesenchyme, and it continues until postnatal day 21. Multiple factors, including growth factor signaling and cell-extracellular matrix (ECM) interactions, are required for normal

[1]Division of Nephrology and Hypertension, Department of Medicine, Vanderbilt University Medical Center, Nashville, TN 37232, USA. [2]Department of Cell and Developmental Biology, Vanderbilt University, Nashville, TN 37240, USA. [3]Research and Medical Services, United States Department of Veterans Affairs, Tennessee Valley Healthcare System, Nashville, TN 37212, USA. [4]Department of Pharmaceutical Sciences, School of Pharmacy, North Dakota State University, Fargo, ND 58108, USA. [5]Department of Molecular Biology, University of Texas Southwestern Medical Center, Dallas, TX 75390, USA. [6]Department of Molecular Physiology and Biophysics, Vanderbilt University, Nashville, TN 37232, USA.
*These authors contributed equally as first authors

‡Author for correspondence (roy.zent@vumc.org)

F.B., 0000-0002-0788-2945; A.T., 0000-0003-1498-9043; J.P.A., 0009-0004-1113-1695; R.Z., 0000-0003-2983-8133

UB branching morphogenesis (Bock et al., 2025; Mathew et al., 2012a).

The principal mediators of cell-ECM interactions are the heterodimeric transmembrane receptors integrins. Integrins function by binding to ECM and by interacting with intracellular proteins that act as mechanical links to the cytoskeleton or signal transduction molecules (Moser et al., 2009). Among the 24 mammalian integrins composed of αβ heterodimers, β1-containing integrins play the most crucial role in kidney collecting system development. Deletion of this integrin subunit at the initiation of UB branching results in a severe morphogenesis defect because integrins are crucial adhesion molecules and are required for the transduction of growth-factor-dependent signals (Zhang et al., 2009). Similarly, inhibiting β1 integrin function by deleting key integrin-binding proteins leads to defective UB development (Bulus et al., 2021; Lange et al., 2009; Mathew et al., 2012b, 2017; Smeeton et al., 2010). Talins, one of the best-studied integrin adaptor proteins, are crucial for UB development in part due to mechanisms independent of β1 integrin binding (Mathew et al., 2017). However, the role of another major adaptor protein family, kindlins, is unexplored in UB development.

The kindlin family of proteins is made up of kindlin 1, kindlin 2 and kindlin 3 (Ussar et al., 2006). Kindlin 1 is found primarily in epithelial cells, kindlin 2 is ubiquitously expressed and kindlin 3 is only expressed in hematopoietic cells (Ussar et al., 2006). All kindlins are structurally similar, with a FERM domain consisting of phosphotyrosine binding (PTB) folds that directly bind to most β integrin subunits (Rognoni et al., 2016). The binding sites of kindlins to the β integrin cytoplasmic tails include the membrane-distal NxxY motif and a highly conserved threonine/threonine motif found in close proximity to the NxxY motif in the mouse β1 integrin cytoplasmic tail (Moser et al., 2008). Mutations in either of these sites disrupts kindlin binding to the integrin cytoplasmic tail (Li et al., 2017). Kindlin 1 and kindlin 2 are expressed early in UB development and persist in the kidney collecting system (Ussar et al., 2006). Although previous studies have shown that both kindlins are essential for normal integrin function in cells (Theodosiou et al., 2016), it is unknown what the role of the kindlin-β1 integrin complex is in organ morphogenesis *in vivo*.

In this study, we sought to understand the role of kindlins in the developing UB and define whether their function is meditated only by binding to β1 integrins. To do this, we simultaneously deleted both kindlin 1 and 2 in the developing UB and compared their phenotypes to mice carrying a Y795A mutation within the distal NPXY motif or the double T788A/T789A mutation in the conserved threonine/threonine motif within the mouse β1 integrin cytoplasmic tail. We also developed kindlin1/2-null CD cells, and β1 Y795A- and T788A/T789A-expressing CD cells to define the cellular phenotypes and signaling defects that underpin the developmental abnormalities observed in these mice. We show that, while kindlins regulate integrin function in kidney collecting system development, they primarily

function as mediators of growth factor signaling required for initial UB formation.

## RESULTS

### Y795A and T788A/T789A mutations in the β1 integrin cytoplasmic tail cause kidney collecting system hypoplasia with epithelial damage

To define the role of β1 integrin kindlin interactions in UB development, we generated mice that selectively carry the Y795A (Y/A) or the T788A/T789A (TT/AA) substitutions in the developing UB. This was achieved by intercrossing mice with heterozygous Y/A or TT/AA mutations with floxed β1 integrin mice and transgenic mice expressing cre recombinase under the Hoxb7 promoter (Bottcher et al., 2012; Mathew et al., 2012b, 2017; Meves et al., 2013; Yu et al., 2002). The offspring had deletion of the floxed β1 allele starting at E10.5 in the UB and expressed only the mutant alleles. The HoxB7-cre; Itgβ1$^{fl/wt}$ mice do not have any obvious

phenotype. As controls, we utilized Itgβ1$^{fl/wt}$/Y/A or Itgβ1$^{fl/wt}$/TT/AA mice, which also do not exhibit a phenotype. The Y/A and TT/AA mutant mice were born in a normal Mendelian ratio, but as they aged, they developed evidence of failure to thrive, and both genotypes died at ∼7 months of age. The kidneys of both the Y/A and TT/AA mice were about 20% smaller than controls when the mice were sacrificed at ∼3 months of age. There was equal expression of β1 integrin in the tubules of the papilla of the control, and Y/A and TT/AA mutants (Fig. S1). Hematoxylin and Eosin staining showed that the papilla was hypoplastic (Fig. 1A). The mutant kidneys had areas of dilated and disorganized fibrosed cortical collecting ducts with a tubulointerstitial infiltration of leukocytes, especially within the medullary rays (Fig. 1A). These features are indicative of severe epithelial damage. The fibrosis in Y/A and TT/AA mice was verified utilizing Picrosirius Red staining and it covered a significant area of the kidneys (Fig. 1B,D). In addition, there was evidence of increased apoptosis measured by TUNEL in the dilated CDs of the Y/A and

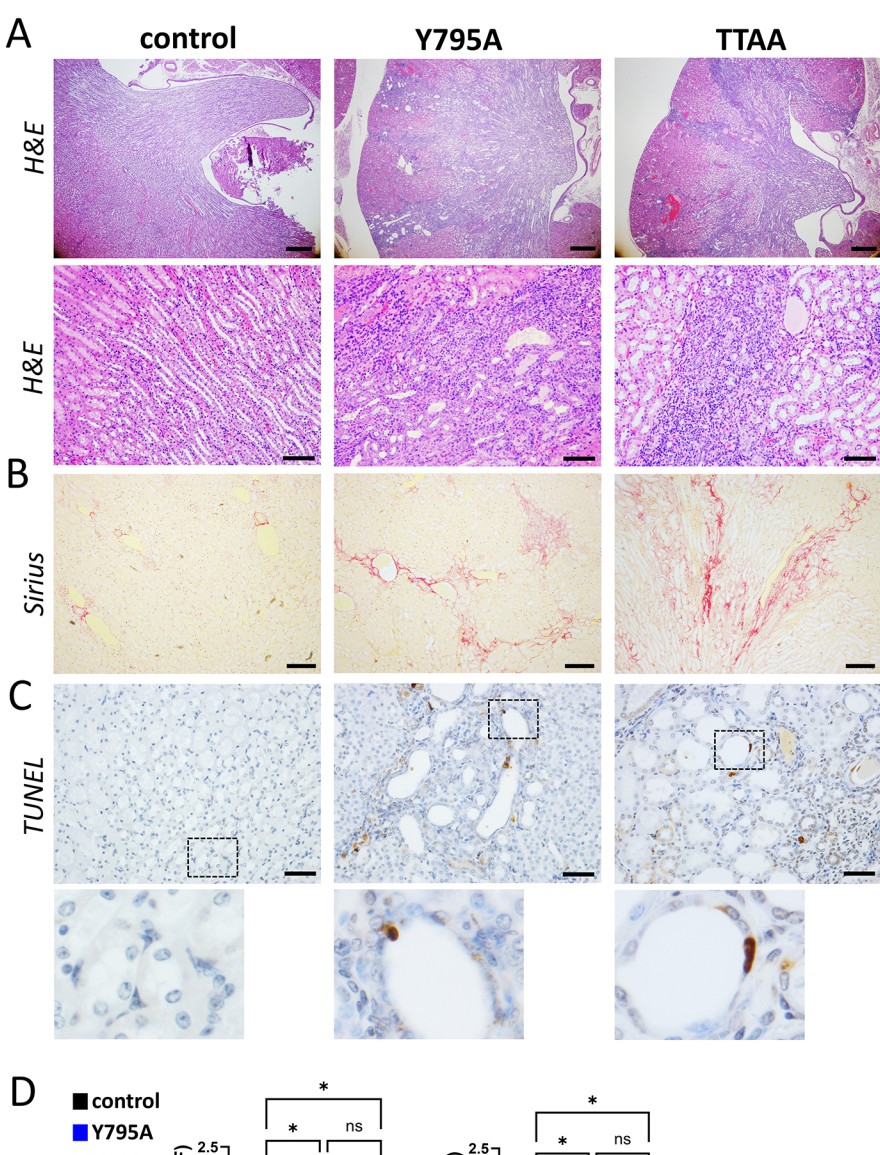

Fig. 1. Y795A and TTAA kidneys show a hypomorphic collecting system with epithelial damage. (A) Kidneys of 3-month-old control, Y795A and TTAA mice stained with Hematoxylin and Eosin (H&E) were visualized by light microscopy. Low- and high-power images are in the top and bottom panels, respectively. (B,D) Kidneys of control and mutant mice were stained with Picrosirius Red (B) and the positive areas per high power field (HPF) were calculated and are shown graphically in D. (C) Sections of kidneys were stained for TUNEL and visualized by light microscopy. The top panels are a low-powered views; the lower panels show high-powered views of a TUNEL-positive apoptotic cells. (D) The number of fibrosis- and TUNEL-positive cells per high-powered field. $n$=4 mice/group. *$P$<0.05; ns, not significant (ANOVA with post-hoc Tukey's test). Data are mean±s.d. Scale bars: 200 µm (A, top; B); 100 µm (A, bottom); 50 µm (C).

TT/AA mice when compared with controls (Fig. 1C,D). Thus, introducing point mutations into the β1 integrin tail that disrupt kindlin binding (Meves et al., 2009, 2013; Montanez et al., 2008; Ussar et al., 2008) causes medullary hypoplasia characterized by tubular epithelial injury, fibrosis and apoptosis.

### Y795A and T788A/T789A mutations in the β1 integrin cytoplasmic tail caused a mild ureteric bud branching morphogenesis defect

We performed histological examinations at various time points to define when the abnormalities in the kidneys were first evident in the Y/A and TT/AA kidneys. Newborn kidneys were ~10% smaller than the wild-type kidneys; however, no overt pathology was observed (Fig. 2A). The medullary regions and papilla appeared smaller in the mutant kidneys and there were less aquaporin 2-expressing collecting ducts (CD) that form from the UB in the mutant kidneys (Fig. 2B,C). These data suggest a mild to moderate branching defect. To definitively identify a developmental UB branching defect in the developing kidneys, we quantified the number of UB tips in intact whole-mounts and optically cleared embryonic kidneys at E15.5 that were stained with antibodies against the UB marker E-cadherin. This was carried out by performing image analysis of the ureteric trees with filament tracing algorithms in Imaris software of images taken by confocal microscopy (Fig. 2D). The numbers of ureteric nodes and tips were reduced by about 30% in the Y/A and TTAA mice when compared to controls (Fig. 2E). Thus, disrupting β1 integrin-kindlin interactions results in a moderate UB branching defect.

### Y795A and T788A/T789A mutations in the β1 integrin cytoplasmic tail cause CD cell dysfunction

To identify the mechanisms of the defects in the Y/A and TT/AA mutants in the β1 integrin subunit, we introduced the same mutations into β1-null CD cells. The cells were sorted so that there was equal surface expression of the wild-type and mutant β1 integrins (Fig. S1). There was no difference in surface expression of integrin β4, α1, α2, α3 and α6 subunits. When the cells were placed in 3D collagen/Matrigel gels, cells expressing the wild-type β1 integrin subunit formed branched tubules, while cells expressing the Y/A or the TT/AA mutant integrin grew as spheroid-like structures with only small branches (Fig. 3A). We next defined which cell functions required for tubulogenesis were affected by these mutations. When the cells were plated on Matrigel there was a significant decrease in adhesion for both the Y/A and TT/AA mutants when compared to control cells (Fig. 3B). Similarly, they migrated significantly less towards Matrigel compared to control cells (Fig. 3C). The Y/A and TT/AA mutants also had a spreading defect (Fig. 3D,F) and formed significantly fewer focal adhesions than control cells when plated on Matrigel (Fig. 3E,G). Surprisingly, there was no significant difference in cell proliferation between Y/A and control CD cells grown on Matrigel, and only a minor difference in proliferation between control or Y/A and TT/AA CD cells (Fig. 3H). Finally, when we investigated whether the mutations altered CD cell polarization, we found a significant decrease in the polarization index of both the Y/A and TT/AA CD cells compared to the wild type, as determined by the ratio of apical to total signal of ZO-1 in CD cells grown on transwells (Fig. 3I,J). These results suggest that inducing a Y/A or TT/AA mutation in the β1 integrin cytoplasmic tail inhibits CD cell branching morphogenesis by inducing mild to moderate adhesion, migration and polarization defects.

Integrins mediate many of their cellular functions by activating well-defined signaling pathways following adhesion to ECM. We

therefore assessed the effects of the Y/A and TT/AA mutations in signaling by CD cells replated on Matrigel-coated plates. Wild-type and mutant cells showed similar time-dependent phosphorylation of FAK, paxillin, Akt, Erk1/2 and p38 (Fig. 4A,B). Matrigel is a heterogeneous substance composed of multiple ECM components such as laminin 111, nidogen, collagen 1 and heparan sulfate proteoglycans, as well as growth factor contaminants like TGF-β and epidermal growth factor, making it difficult to define the specificity of integrin-dependent signaling (Kleinman and Martin, 2005). We therefore performed the same experiment on purified collagen 1 and laminin 511 (ligands for β1 integrins), and vitronectin, which is a ligand for αv integrins. We also see similar time-dependent phosphorylation of FAK, paxillin, Akt, Erk1/2 and p38 in wild-type and mutant cells on these substrates (Fig. S2). GDNF (glial cell line-derived neurotrophic factor) is the crucial growth factor for UB branching (Costantini, 2010) and it requires intact integrin signaling to mediate its effects (Zhang et al., 2009). Wild-type and mutant cells showed similar response to GDNF-induced activation of intracellular signaling (Fig. 4C,D). Thus, mutations that inhibit kindlin binding to the β1 integrin subunit have a marginal effect on integrin-dependent signaling following CD cell adhesion or after exposure to growth factors.

### Kindlins are required for kidney collecting system development

We show that mutating the kindlin-binding sites on the β1 integrin cytoplasmic tail causes a moderate UB branching morphogenesis phenotype and subsequent formation of a hypoplastic collecting system that undergoes fibrosis as the mice age. We next defined whether this phenotype is dependent on the interaction of kindlins with the β1 integrin subunit rather than β1 integrin-independent functions of kindlin. To do this, we deleted kindlin 1 and kindlin 2 in the developing UB by crossing the kindlin 1 and kindlin 2 floxed mice, and subsequently breeding these double homozygotes (K1$^{fl/fl}$:K2$^{fl/0}$) with Hoxb7-cre mice (Hoxb7:K1$^{fl/fl}$:K2$^{fl/0}$). This breeding strategy resulted in no live mice with the Hoxb7:K1$^{fl/fl}$:K2$^{fl/0}$ genotype and 100% of these newborn mice were anephric (Fig. 5A). To determine when kidney development was arrested in the mutants, we initially looked at E15.5 kidneys in Hematoxylin and Eosin-stained paraffin wax-embedded kidney sections and 3D reconstructions of E-cadherin-stained retroperitoneums (Fig. 5B,C). There was no evidence of kidneys, suggesting that there was either a defect in the ability of the UB to develop from the Wolffian duct or a failure of UB development after it interacts with the metanephric mesenchyme. To investigate these possibilities, we dissected Wolffian ducts from E10.5 embryos and cultured them ex vivo for 12 h (Fig. 5D,E). In control mice, a widened Wolffian duct was seen at E10.5 and a UB that had grown into the metanephric mesenchyme was visualized after 12 h in culture (Fig. 5F). In contrast, there was no Wolffian duct widening or UB budding in the Hoxb7:K1$^{fl/fl}$:K2$^{fl/0}$ kidneys. This result suggests that kindlins are required for the initiation of UB growth out of the Wolffian duct, which is a process that is highly dependent on growth factor dependent signaling.

### Kindlins are required for CD cell tubulogenesis

To define the mechanisms underlying kidney agenesis in the Hoxb7:K1$^{fl/fl}$:K2$^{fl/0}$ mice, we derived CD cells from kindlin-1$^{fl/fl}$ kindlin-2$^{fl/fl}$ mice and deleted the genes in vitro using adeno-cre virus to generate kindlin 1/2 null (K1/K2$^{-/-}$) CD cells (Fig. 6A). Mock transfected Kindlin-1$^{fl/fl}$/Kindlin-2$^{fl/fl}$ CD cells without adeno-cre were used as controls. K1/K2$^{-/-}$ CD cells lost their

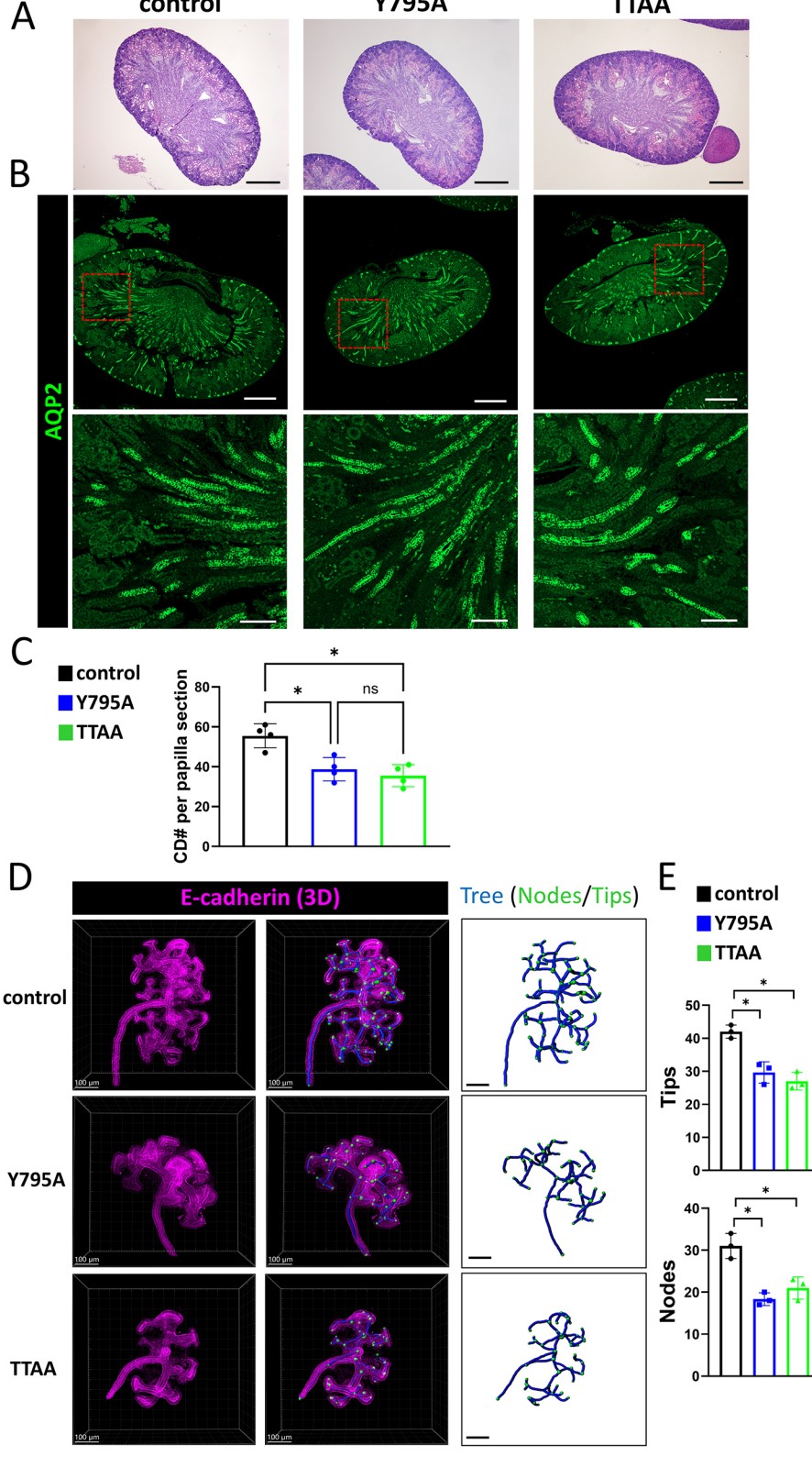

**Fig. 2. Y795A and TTAA kidneys have moderate UB branching morphogenesis defects.**
(A) Kidneys from newborn control, Y795A and TTAA mice were stained with Hematoxylin and Eosin and visualized by light microscopy.
(B) Kidneys from newborn mice stained using antibodies directed against aquaporin 2. The top panels demonstrate a cross-section through a whole kidney and the bottom panels are high-power magnifications of the medullary collecting ducts (outlined in the top panels). (C) The number of collecting ducts per papillary section in B was quantified and is represented graphically. $n$=4 mice per group. (D) Three-dimensional branching tree reconstructions (Imaris) of the E-cadherin-labeled UB (magenta) after optical clearing at embryonic stage E13.5. The center column shows the branching tree overlaid with filament tracing (blue lines). Nodes and tips are shown as green dots. The far-right column outlines the branching structure without E-cadherin. (E) Quantification of ureteric bud tree tips and nodes shown in D ($n$=3 mice per group). *$P$<0.05; ns, not significant (ANOVA with post-hoc Tukey's test). Data are mean±s.d. Scale bars: 500 μm (A); 400 μm (B, top); 100 μm (B, bottom; D).

ability to adhere to a tissue culture dish and grew extremely slowly as free-floating cells. We confirmed viability of the K1/K2$^{-/-}$ CD cells by staining live cells with Hoechst 3342 and performing a Trypan Blue exclusion assay (Fig. S3). The K1/K2$^{-/-}$ and control CD cells did not display major differences in cell surface expression of the β1, α1, α2, α3 and α6 integrin subunits. When they were placed in a three-dimensional collagen/Matrigel gel, K1/K2$^{-/-}$ CD cells were unable to form a tubule or branch, and failed to form a morphologically intact epithelial structure, suggesting a severe tubulogenesis defect (Fig. 6B). As predicted, the K1/K2$^{-/-}$ CD cells adhered and migrated very poorly on Matrigel (Fig. 6C,D). Consistent with these defects, K1/K2$^{-/-}$ CD cells failed to spread

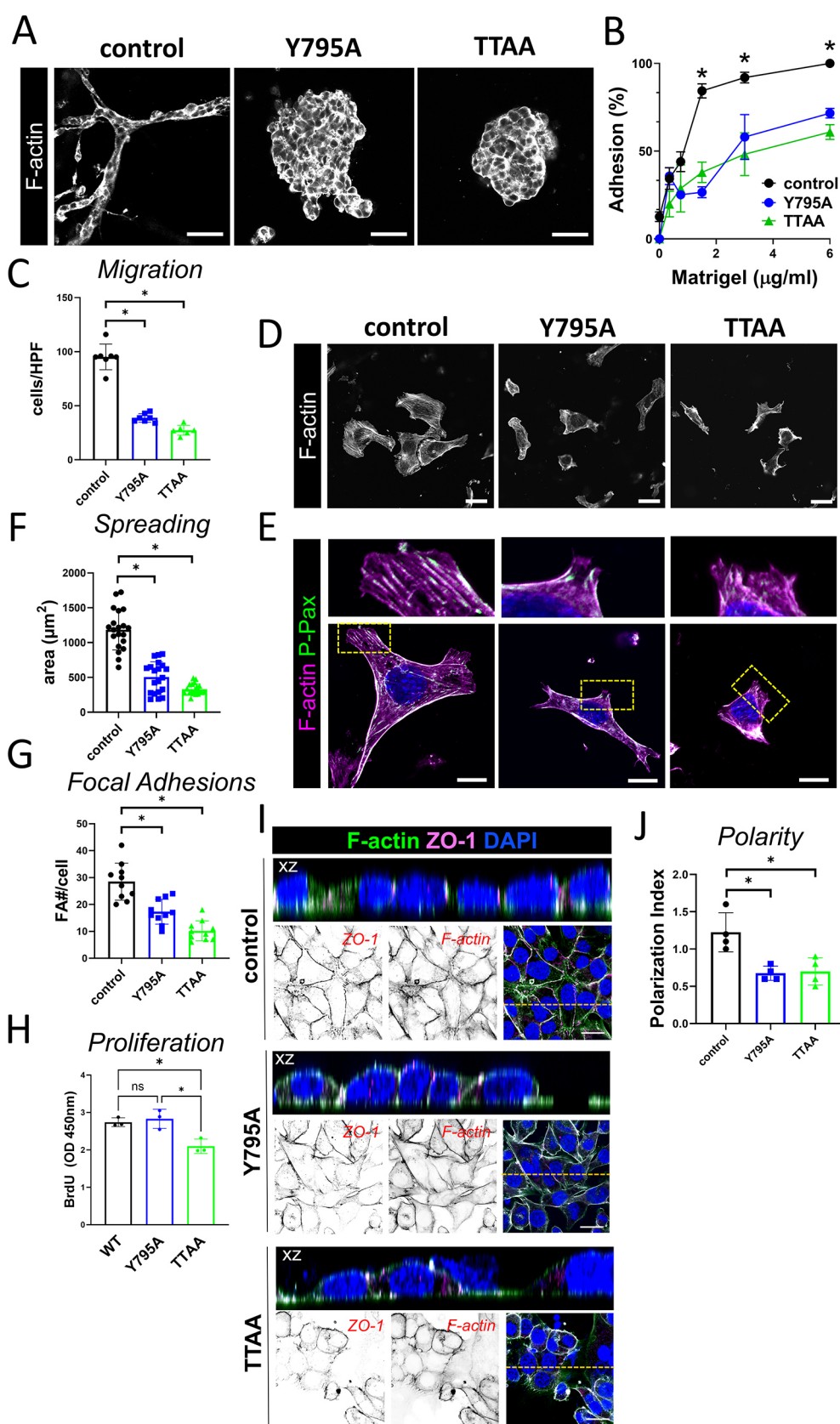

**Fig. 3. Y795A and TTAA CD cells have defective integrin-dependent functions.** (A) Control, Y795A and TTAA mutant CD cells were placed in 3D collagen I/Matrigel gels for 7 days, stained with rhodamine-phalloidin and visualized by confocal microscopy. (B) CD cell populations were allowed to adhere to Matrigel at the indicated concentrations for 1 h. Percentage adhesion is demonstrated on the *y*-axis and the concentration of Matrigel is on the *x*-axis (*n*=3 experiments). Significance (*$P$<0.05) is indicated between control and Y795A or TTAA at the indicated concentration (unpaired *t*-test). (C) CD cells were plated on transwells coated with Matrigel and the number of cells that had migrated after 4 h was counted and expressed as cells per high powered field (HPF). *n*≥6 HPFs pooled from three experiments. (D) CD cells were plated and allowed to spread for 1 h on Matrigel, stained with rhodamine-phalloidin (F-actin) and visualized. (E) CD cells were plated on Matrigel after which they were stained with rhodamine phalloidin and antibodies directed against phospho-paxillin (P-Pax, green), and visualized with confocal microscopy. Images are representative of *n*=3 repeat experiments. (F) Quantification of spreading area per cell of at least 20 cells per group. (G) Quantification of focal adhesion (FA) number per cell of at least 10 cells per group. (H) CD cell proliferation, as assessed by BrdU incorporation, 1 h after plating on Matrigel colorimetrically (OD 450 nm). The averages of three experiments are shown. (I) CD cells were grown to confluence as monolayers on transwells, and stained for F-actin (rhodamine phalloidin green), ZO-1 (tight junctions, magenta) and nuclei (DAPI, blue). Shown are *xz*-orthogonal projections sliced at the yellow dotted line of the *z*-stack cross-section. Single-channel images of ZO-1 and F-actin are shown as inverted grayscale images. (J) Quantification of polarity by polarization index (ratio of apical over total ZO-1 immunofluorescence). Averages of four monolayers per group are shown. Images are representative of three samples per group from 3 independent replicates. *$P$<0.05; ns, not significant (ANOVA with post-hoc Tukey's test). Data are mean±s.d. Scale bars: 40 µm (A); 20 µm (D,I); 10 µm (E).

(Fig. 6E,G) and to form stress fibers or focal adhesions, as defined by phospho-paxillin staining (Fig. 6F,H). In addition, the K1/K2$^{-/-}$ CD cells had a severe proliferation defect when they were grown on Matrigel (Fig. 6I). These cells were also unable to form polarized monolayers when grown on transwells (Fig. 6J), resulting in a polarization index that was close to zero when calculated from ZO-1 staining (Fig. 6K). The K1/K2$^{-/-}$ CD cells lost E-cadherin expression but they overexpressed ZO-1 relative to

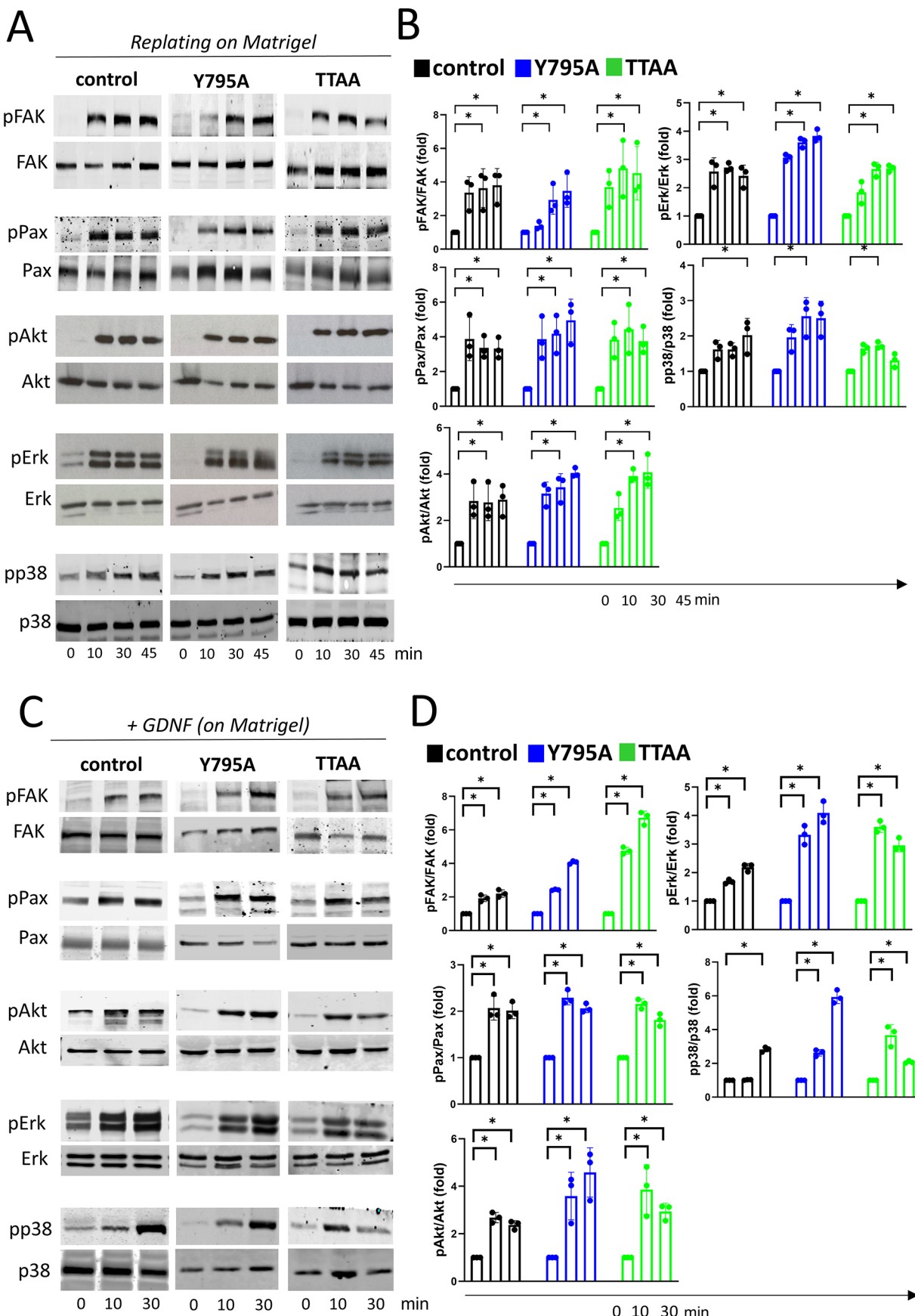

**Fig. 4.** See next page for legend.

**Fig. 4. Integrin β1 Y795A or TTAA mutations do not significantly alter integrin-dependent signaling.** (A,B) Control, Y795A and TTAA CD cells were plated on Matrigel in serum-free medium. (A) Cell lysates were analyzed after replating at the indicated time points by western blotting for levels of phosphorylated and total focal-adhesion kinase (FAK), paxillin (Pax), Akt, Erk and p38. (B) Densitometry of the bands of the phosphorylated and total proteins was carried out using Image J and the ratios are represented graphically. The fold changes in phosphorylated/total protein at different time points relative to the '0' time point (which was set at fold change=1) are shown. (C,D) CD cells were allowed to adhere to Matrigel overnight, after which they were treated with GDNF for the times indicated. The cells were then lysed and analyzed by western blotting for the phosphorylated and total protein forms of the indicated proteins. (D) Levels of phosphorylated proteins were measured by densitometry and normalized to total protein, as indicated above. Blots are representative of three experiments. *$P<0.05$; ns, not significant (ANOVA with post-hoc Tukey's test). Data are mean±s.d.

control CD cells (Fig. 6L). Thus, deleting kindlin 1 and 2 in CD cells causes severe defects in adhesion, migration, polarization and proliferation, which results in their inability to undergo tubulogenesis *in vitro*.

### Kindlins are crucial for adhesion- and growth factor-dependent signaling

As there were major adhesion defects in the K1/K2$^{-/-}$ CD cells, we investigated their signaling in response to plating on ECM. Plating control cells onto Matrigel-coated plates (Fig. 7A,B) or on collagen 1, laminin 511 and vitronectin (Fig. S4) resulted in phosphorylation of FAK and paxillin, as well as their downstream effector proteins Akt, Erk 1/2 and p38 in a time-dependent manner. As expected, the K1/K2$^{-/-}$ CD cells were unable to phosphorylate FAK or paxillin at any time after plating. Surprisingly, there were no differences in Akt, Erk 1/2 or p38 activation between the control and K1/K2$^{-/-}$ CD cells, suggesting that activation of these pathways does not require kindlins following cell adhesion to ECM (Fig. 7A,B).

When control CD cells were placed on Matrigel and then treated with GDNF, there was robust phosphorylation of FAK, paxillin, Akt, Erk 1/2 and p38 within 10 min. In contrast, there was no activation of any of these pathways in the K1/K2$^{-/-}$ CD cells (Fig. 7C,D). This result suggests that kindlins are required for GDNF-mediated signaling in CD cells. To investigate whether the diminished signaling in response to GDNF in K1/K2$^{-/-}$ CD cells was dependent on integrin adhesion to ECM, we performed the same experiment with the cells plated on plastic. Cells plated on plastic showed decreased GDNF-induced signaling, suggesting that kindlins function in an integrin-independent manner (Fig. S5). To verify the physiological importance of this finding, we performed an ectopic budding assay *ex vivo*. We isolated the Wolffian ducts from control and Hoxb7:K1$^{fl/fl}$:K2$^{fl/0}$ E10.5 embryos, and cultured them *ex vivo* for 48 h with 100 ng/ml GDNF. Two or three ectopic buds grew out of the control Wolffian ducts, while no budding was seen from the kindlin double-KO buds (Fig. 7E,F). These results suggest that, in addition to their role in integrin function, kindlins are required for GDNF-dependent signaling and early UB formation.

It is possible that kindlins induce their effects via non β1 integrins. To investigate this, we generated Hoxb7:αv$^{fl/fl}$ mice (lacking αvβ3, αvβ5, αvβ6 or αvβ8), which were completely normal. We therefore attempted to generate Hoxb7:Itg αv$^{fl/fl}$/β1$^{fl/fl}$ mice, which lack all the major integrin receptors found in the UB. Our breeding strategy resulted in 92 newborn mice, of which 41 (45%) did not express cre and were used as control mice. Of the cre

positive mice, 36 (39%) were Hoxb7:Itg αv$^{fl/+}$/β1$^{fl/+}$mice, 14 (15%) were Hoxb7:Itg αv$^{fl/+}$/β1$^{fl/fl}$ mice and only 1 (1%) was a Hoxb7:Itg αv$^{fl/fl}$/β1$^{fl/fl}$ mouse. The Hoxb7:Itg αv$^{fl/fl}$/β1$^{fl/fl}$ mouse was anephric and we verified that all the Hoxb7:αv$^{fl/fl}$/β1$^{fl/fl}$ embryos we dissected at E15.5 were also anephric. Twelve of the 14 (86%) Hoxb7:αv$^{+/fl}$/β1$^{fl/fl}$ mice were anephric and only two (14%) developed single kidneys, which had major UB branching defects that were worse than the Hoxb7:β1$^{fl/fl}$ mice (Zhang et al., 2009) (Fig. S6). There was no phenotype in any of the 21 Hoxb7:Itg αv$^{fl/+}$/β1$^{fl/+}$ mice (Fig. S6). We generated αv$^{-/-}$/β1$^{-/-}$ CD cells but no mechanistic studies were performed on them as they were unable to adhere to any ECM substrates. Thus, we can conclude that heterozygosity of integrin β1 and αv is sufficient for normal UB development, and that αvβ3, αvβ5, αvβ6 or αvβ8 integrins play little or no role in UB development when at least one copy of β1 integrin is expressed.

## DISCUSSION

The goal of this study was to understand the role of kindlins in UB branching morphogenesis and whether their function is dependent or independent of their interactions with integrin β1. We demonstrate that disrupting the binding of kindlins to the β1 integrin cytoplasmic tail result in a moderate UB developmental abnormality. By contrast, co-deletion of both kindlin isoforms expressed resulted in a failure of the UB to develop from the Wolffian duct due to a major adhesion defect and its inability to respond to GDNF. Kindlin null CD cells also had a major adhesion defect and were unable to activate GDNF-dependent signaling, irrespective of integrin-dependent adhesion. Thus, in addition to regulating integrin functions, kindlins mediate the β1 integrin-independent GDNF signaling required for the initiation of UB development.

The two NxxY motifs of the β1 integrin tail are key binding sites for integrin binding proteins (Calderwood et al., 1999; Moser et al., 2008, 2009). Talins bind to the membrane proximal NxxY motif (Wegener et al., 2007), while kindlins bind to the membrane distal NxxY and adjacent TT motifs via highly conserved amino acids in the kindlin FERM domain (Moser et al., 2008, 2009). Germline Y/A mutations in one or both NxxY motifs cause preimplantation mortality in mice (Bottcher et al., 2012; Meves et al., 2013). Expression of the same mutations in both NxxY motifs in the epidermis of mice caused a phenotype similar to that seen in mice lacking the integrin β1 integrin subunit in the epidermis; however, only minor abnormalities occurred when mutations in the single motifs were introduced (Meves et al., 2013). Unlike the skin, expression of Y/A mutations in both NxxY motifs in the developing UB had a less severe phenotype than that of mice lacking β1 in the UB with a moderate branching morphogenesis phenotype followed by tubular destruction and severe renal fibrosis (Mathew et al., 2012b). When a Y783A mutation (membrane proximal NxxY motif) was introduced into the UB, the phenotype was less severe than the double YY/AA mutant, and it resulted in a mild to moderate branching morphogenesis abnormality followed by renal fibrosis (Mathew et al., 2017). This defect was very similar to that seen in both the Y795A and T788A/T789A mutant mice analyzed in this study, which is consistent with both these motifs being crucial for kindlin binding to the β1 integrin cytoplasmic tail and with the developmental phenotype being due to the loss of kindlin binding. It also concurs with a study in which T788A/T789A and Y795A germline mutations resulted in reabsorbed embryos at E7.5 (Bottcher et al., 2012). Thus, although introducing mutations into the membrane proximal or distal NxxY motifs, or into the TT motif adjacent to the distal NxxY motif totally abrogates talin

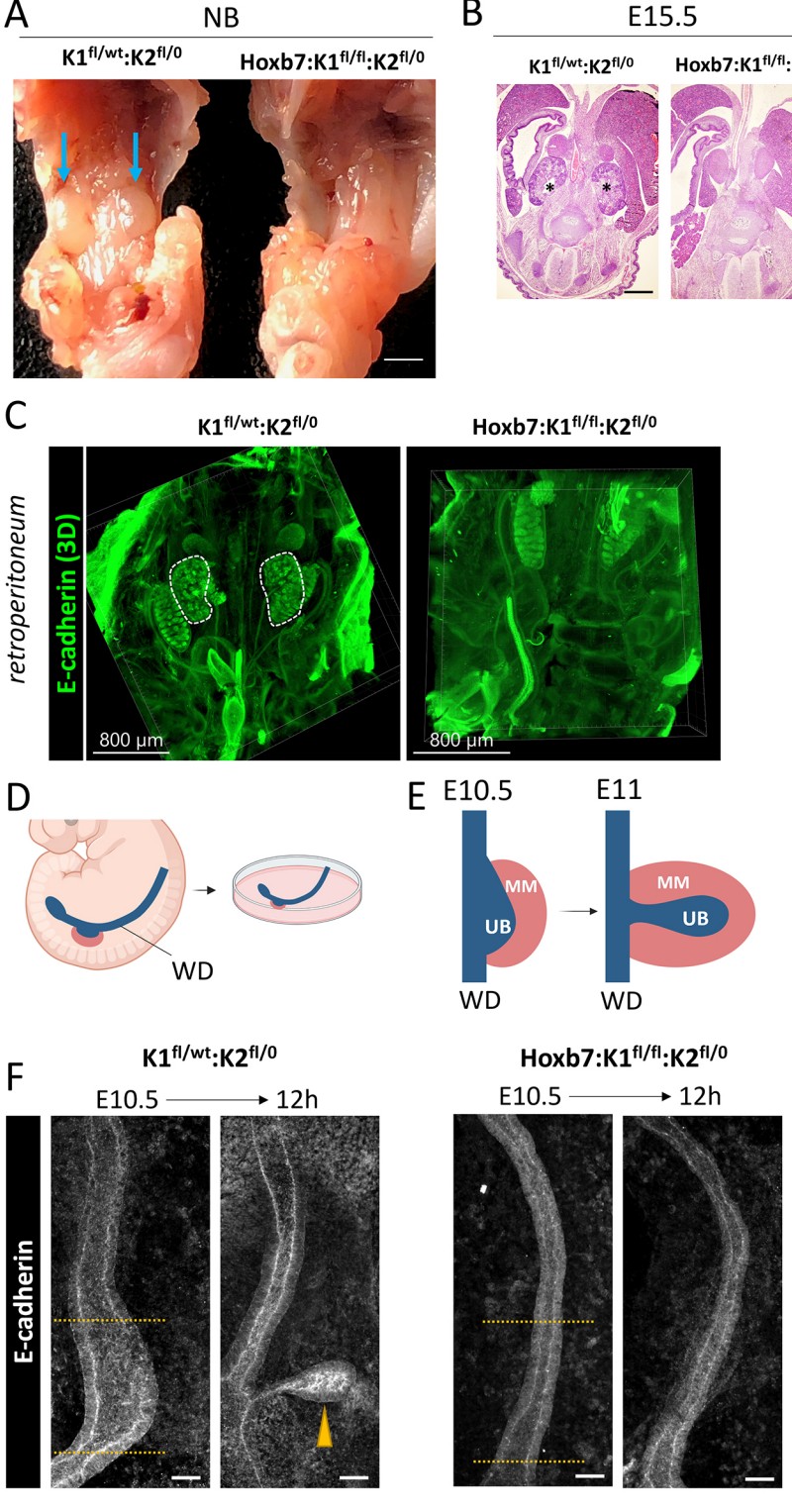

**Fig. 5. UB-specific kindlin 1/2 double-null mice do not develop kidneys.** (A) Photographs of the retroperitoneal space of newborn control (Kindlin 1$^{fl/wt}$: Kindlin 2$^{fl/0}$) and kindlin null (Hoxb7:Kindlin 1$^{fl/fl}$: Kindlin 2$^{fl/0}$) mice. Blue arrows indicate kidneys in control mice. (B) Retroperitoneal paraffin wax-embedded cross-sections of embryos at stage E15.5 stained with Hematoxylin and Eosin. Kidneys are indicated with asterisks in the control mice. (C) Full three-dimensional (3D) reconstructions of the retroperitoneal kidney space of E15.5 mice stained with the epithelial UB-marker E-cadherin (green). The developing kidneys are outlined by white dotted lines. (D,E) Schematics depicting the wolffian duct (WD) isolation (D) followed by culture and UB budding (E). (F) Wolffian ducts (labeled with E-cadherin) isolated at E10.5 and visualized by confocal microscopy after optical clearing before and 12 h after culture of control and kindlin double-KO UBs. The thickened WD region where UB budding is expected is outlined by yellow lines and a yellow arrowhead. Images are representative of two experiments. Scale bars: 2 mm (A); 200 μm (B); 800 μm (C); 40 μm (F). Created in BioRender by Bock, F., 2026. https://BioRender.com/u8opvnx. This figure was sublicensed under CC-BY 4.0 terms.

(Mathew et al., 2017) or kindlin binding, they only alter some integrin functions in the setting of UB development.

Deletion of β1 integrin results in major abnormalities in CD cell adhesion, migration, proliferation and growth factor signaling (Zhang et al., 2009). By contrast, Y/A mutations in both NxxY motifs in CD cells decreased cell adhesion and migration, as well as activation of Akt or Erk following growth factor activation by ∼50% (Mathew et al., 2012b). A less severe defect was seen in CD cells where a Y/A mutation was introduced into the β1 membrane proximal NxxY motif (Mathew et al., 2017). We now demonstrate very similar defects with mutations in the distal β1 NxxY motif or the highly conserved TT/AA motif to those seen in CD cells with a membrane proximal mutation. The fact that mutations affecting either talin or kindlin binding alter similar integrin functions is consistent with the ability of both talin and kindlin to induce conformational changes in the β1 integrin tail, and with the finding that talin and kindlin directly interact with each other (Aretz et al., 2023). It also suggests that some functional consequences of

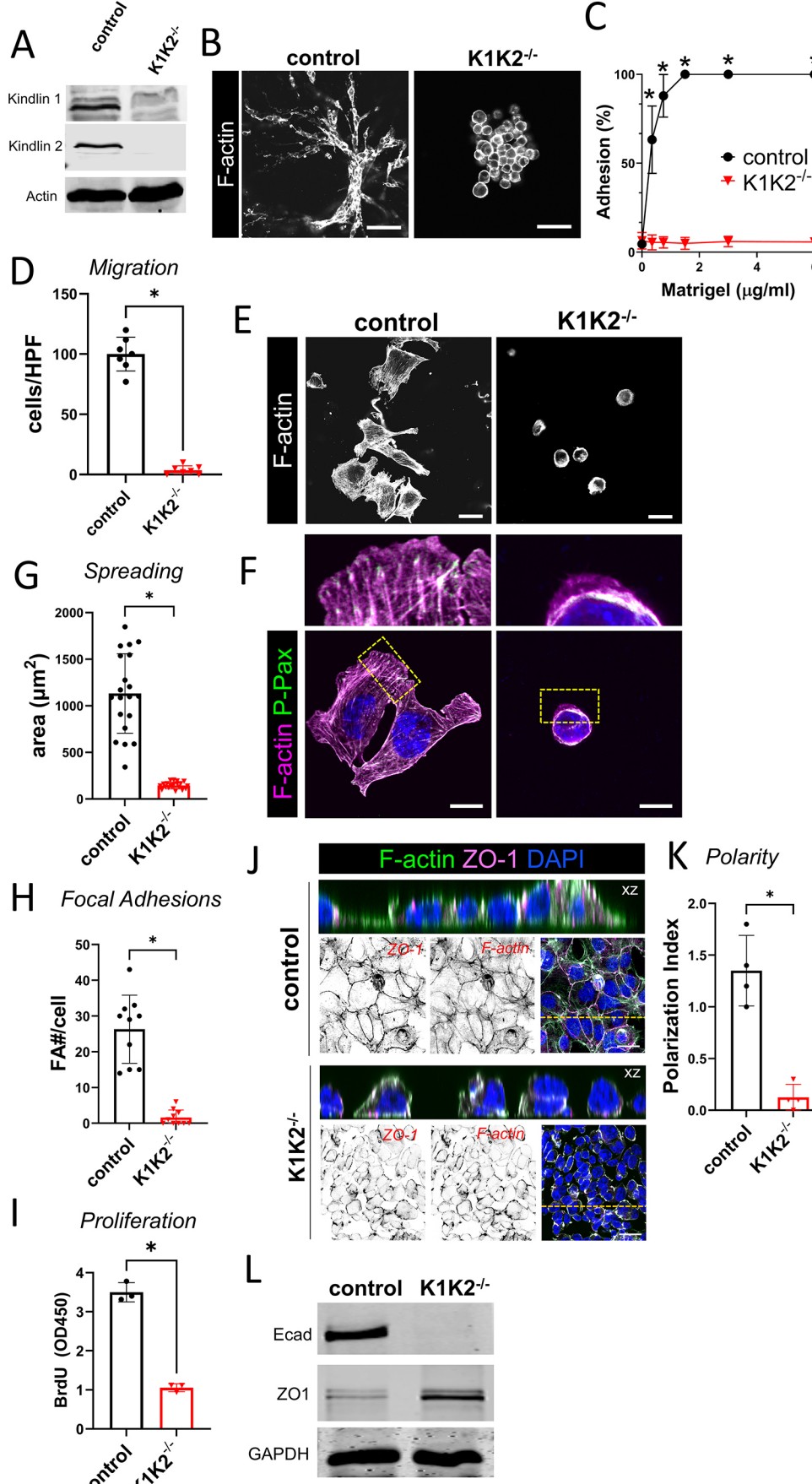

**Fig. 6. Kindlin 1/2 double-null CD cells are unable to adhere, migrate or spread.** (A) Immunoblotting of control and K1K2$^{-/-}$ CD cell total lysates for kindlin 1 and kindlin 2 with actin as the loading control. (B) Control and kindlin 1/2 double-null CD cells (K1K2$^{-/-}$) were placed in 3D collagen I/Matrigel gels for 7 days, stained with rhodamine-phalloidin and visualized by confocal microscopy to assess tubulogenesis. (C) CD cell populations were allowed to adhere to Matrigel at the indicated concentrations for 1 h. $n$=3 experiments. *$P$<0.05 between control and K1K2$^{-/-}$ CD cells at the indicated concentration (unpaired $t$-test). (D) CD cells were plated on transwells coated with Matrigel, and migration, measured as cells per high powered field (HPF), was evaluated after 4 h. $n$≥6 HPFs pooled from three experiments. (E) CD cells were allowed to spread for 1 h on Matrigel and stained with rhodamine-phalloidin (F-actin). (F) After spreading, rhodamine-stained (magenta) cells were co-labeled for the focal-adhesion protein phospho-paxillin (P-Pax, green) and visualized with confocal microscopy. Data are representative of $n$=3 repeat experiments. (G) Quantification of spreading area per cell of at least 20 cells per group. (H) Quantification of focal adhesions (FAs) per cell of at least 10 cells per group. (I) CD cell proliferation, as colorimetrically (OD 450 nm) assessed by BrdU incorporation 1 h after plating on Matrigel. Averages of three experiments are shown. (J) Transwell polarity assay of CD cell monolayers grown to confluence over 24 h in transwell inserts and visualized with confocal microscopy of phalloidin (F-actin, green), ZO-1 (tight junctions, magenta) and nuclei (DAPI, blue). Shown are $xz$-orthogonal projections sliced at the yellow dotted line of the $z$-stack cross-section. Single-channel images of ZO-1 and F-actin are shown as inverted grayscale images. (K) Quantification of polarity by polarization index (ratio of apical over total ZO-1 immunofluorescence). Averages of four monolayers per group are shown. (L) Immunoblotting of CD cell total lysates for E-cadherin (Ecad) and ZO-1 with GAPDH as the loading control. Representative of three samples per group from independent replicates. The control data are reproduced from Fig. 3 as these experiments were performed simultaneously. *$P$<0.05; ns, not significant (unpaired $t$-test). Data are mean±s.d. Scale bars: 40 µm (B); 20 µm (E,J); 10 µm (F).

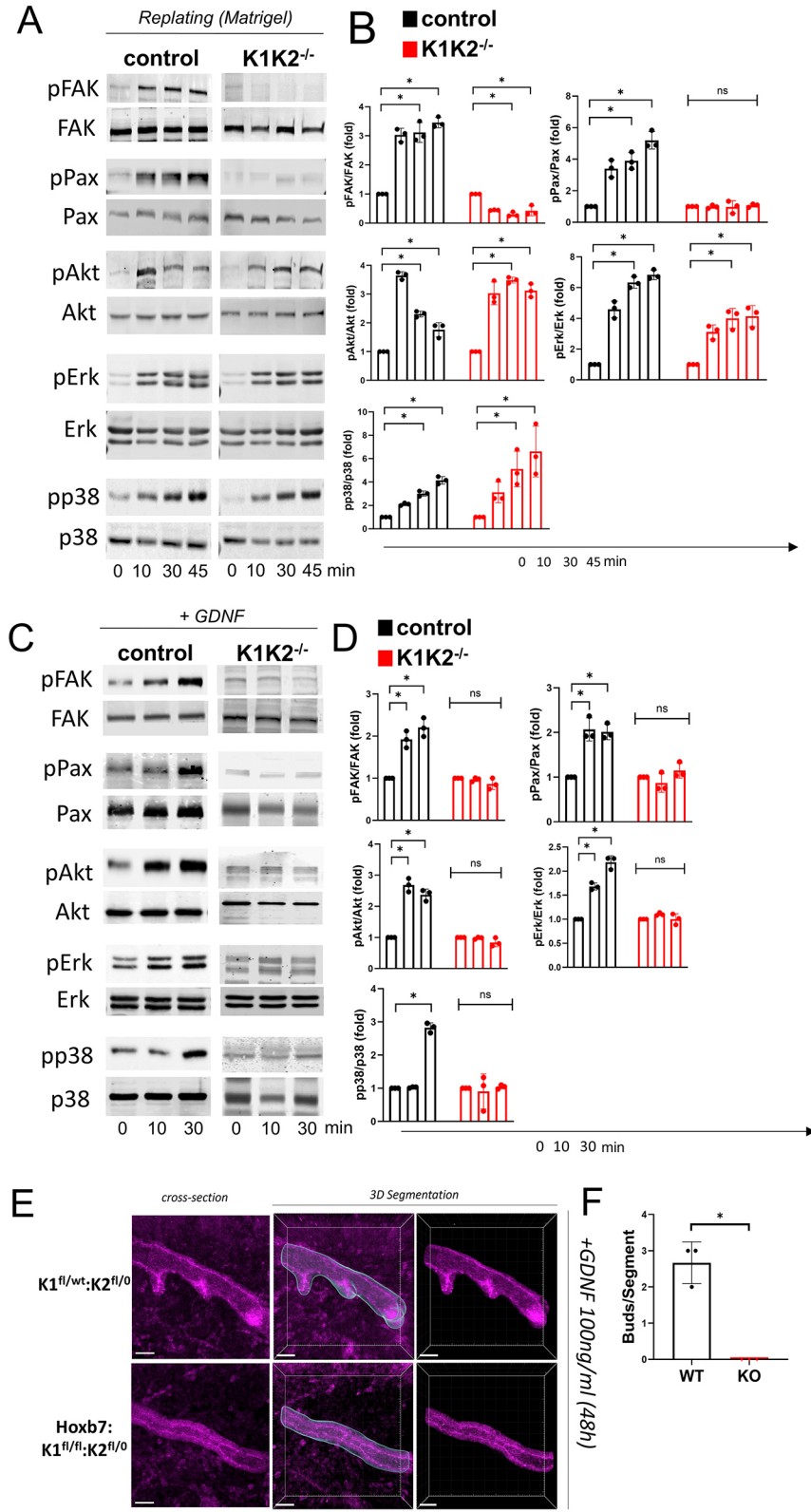

**Fig. 7. Kindlin 1/2 double-null CD cells have integrin signaling defects and are unable to respond to GDNF.** (A,B) Control and K1K2⁻/⁻ CD cells were plated in serum-free medium on Matrigel. Cell lysates were analyzed after replating at the indicated time points by western blotting for levels of phosphorylated and total focal-adhesion kinase (FAK), paxillin (Pax), Akt, Erk and p38. Phosphorylated protein levels were measured by densitometry using Image J and normalized to total protein. The fold changes in phosphorylated/total protein at different time points relative to the '0' time point (which was set at fold change=1) are shown in B. (C,D) CD cells were allowed to adhere to Matrigel overnight, after which they were treated with the growth factor GDNF for the times indicated. The cells were then lysed and analyzed by western blotting for the phosphorylated and total protein forms of the indicated proteins. Levels of phosphorylated proteins were measured by densitometry and normalized to total protein, as indicated above, in D. Images are representative of three experiments. (E,F) *Ex vivo* ectopic budding assay of cultured Wolffian duct explants from control and Hoxb7-Cre Kindlin 1/2 double-null mice treated with 100 ng/ml GDNF for 48 h and quantified in F as buds per segment from three replicates per group. Scale bars: 20 µm. *$P<0.05$; ns, not significant (ANOVA with post-hoc Tukey's for B and D; unpaired *t*-test for F test). Data are mean±s.d.

talin-kindlin interactions are mediated by associations between the integrin β1 tail and either talin or kindlin (Aretz et al., 2023).

Deleting both kindlins results in the inability of the UB to grow out of the Wolffian duct. This extreme phenotype is similar to the deletion of the GDNF receptor, Ret, in the UB (Schuchardt et al., 1994) and is worse than the talin 1/2 deletion in the UB, where the

UB grows into the MM but branching is arrested at day E12.5 (Mathew et al., 2017). The discrepancy between the kindlin and talin mice is somewhat unexpected as both kindlin and talin null CD cells have severe adhesion and migration defects, with the defect being more severe in the talin-deficient cells (Mathew et al., 2017). Integrin-dependent signaling induced by adhesion of the

kindlin-null CD cells demonstrates diminished activation of FAK and paxillin with no effects on Akt, Erk and p38 activation. The defects in FAK and paxillin phosphorylation are similar to those seen in kindlin null fibroblasts and are consistent with the findings that kindlin binds directly to paxillin, which in turn interacts with FAK, resulting in its activation/phosphorylation (Theodosiou et al., 2016). Interestingly, there is no activation of any signaling pathways in kindlin null CD cells in response to GDNF irrespective of whether they are plated on ECM or plastic. While it is possible that the CD cells expressed their own ECM, the signaling experiments were performed over short time frames making it unlikely that the cells produce a matrix that affects signaling. Furthermore, isolated kindlin null UBs did not respond to GDNF in vitro. These data suggest that, in addition to regulating integrin-dependent adhesion via FAK and paxillin, kindlins also regulate GDNF signaling via alternative mechanisms. This may be due to their ability to interact with the Ret receptor per se, as kindlin 2 directly interacts with TGF-β receptor 1, which stabilizes the β1 integrin:TβRI complex and regulates downstream oncogenic signaling (Yousafzai et al., 2024). In addition, kindlin 1 can regulate EGF-dependent cell migration by directly associating with the EGF receptor (Michael et al., 2019).

Our data show that deleting all integrins or both kindlins in the developing UB results in anephric mice, which demonstrates their importance for this developmental process. If kindlins induced all their effects via binding to β1 integrins then the kindlin-binding mutants would have phenocopied the kindlin null mice; however, this was not the case. Our data that Hoxb7:αv$^{fl/fl}$ mice were normal but Hoxb7:αv$^{+/fl}$/β1$^{fl/fl}$ mice were anephric or had major UB branching defects that were significantly worse than those in Hoxb7:β1$^{fl/fl}$ mice (Zhang et al., 2009), establish that the β1 integrins are the most important integrins for UB development and at least one copy of the β1 integrin gene is required. It further suggested that kindlin binding to αv integrins was not important in UB development. Thus, we can conclude that kindlin binding to β1 integrins is required for normal UB development. In addition, our data suggest it is unlikely (but possible) that the only mechanism whereby kindlins or talins mediate UB development is by integrin binding. We propose that kindlins regulate integrin function and GDNF signaling by distinct mechanisms.

To the best of our knowledge, this is the only study where a double deletion of kindlin 1 and 2 has been performed in vivo, and it clearly demonstrates their importance in kidney development. Both kindlins are highly expressed in the kidney so this study defines a class rather than an isoform effect (Ussar et al., 2006). The phenotype was extremely severe and is consistent with the germline deletion of kindlin 2 that results in lethality at E7.5 (Montanez et al., 2008) and with a major platelet phenotype in the kindlin 3 null mice, where kindlin 3 is the only isoform present (Moser et al., 2008). By contrast, germline deletion of kindlin 1 resulted in perinatal mortality from gastrointestinal and skin abnormalities due to aberrations in cell adhesion (Ussar et al., 2008). It has further been shown that deleting kindlin 1 only in the skin recapitulates Kindlers syndrome, where there is an adhesion defect from decreased β1 integrin activation and alterations in the TGFβ/Wnt signaling pathways mediated by integrin β6 cytoplasmic tails (Rognoni et al., 2014). Of note, although both kindlins 1 and 2 are expressed in skin, kindlin 2 could not compensate for all the kindlin 1 functions, suggesting they perform isoform specific functions (Bandyopadhyay et al., 2012). Kindlin 2 has been deleted in multiple organs, causing a variety of phenotypes (Cao et al., 2020; Chen et al., 2022; Chi et al., 2021; Dong et al., 2023; Gao et al., 2019, 2023; He et al., 2020; Krenn et al., 2020; Moretti et al., 2018; Sarvi et al., 2018; Sun et al., 2017; Wang et al.,

2023; Wu et al., 2015; Zhang et al., 2016, 2019). None of these organ specific deletions induced a phenotype as severe as we observed, and initiation of organ development always occurred. The reasons for this are likely that the kindlin isoforms can compensate for different cell functions, including in the developing UB. It is unknown whether kindlin 1 or kindlin 2 is the dominant kindlin required for UB development.

In conclusion, we provide evidence that, in addition to their role in regulating integrin function, kindlins are required for the growth factor signaling necessary for UB formation. This function is not simply mediated by the interactions of kindlins with β1 integrins and is either induced by their interactions with other integrins or, more likely, is due to yet to be determined integrin-independent mechanisms that may include a direct interaction with growth factor receptors.

## MATERIALS AND METHODS
### Generation of mice
Mice with β1-integrin threonine-to-alanine mutation in positions 788 and 789 (β1 TT/AA) and the tyrosine-to-alanine mutation in position 795 (β1 Y795A) have been described previously (Bottcher et al., 2012; Czuchra et al., 2006). These transgenic mice were crossed with Itgβ1$^{fl/fl}$ mice (Brakebusch et al., 2000) and mated with Hoxb7cre mice (Yu et al., 2002) to generate mice with UB restricted point mutations of β1 integrin. To generate Hoxb7:K1$^{fl/fl}$:K2$^{fl/0}$ mice, we crossed floxed kindlin 1 (Rognoni et al., 2014) and kindlin 2 (Theodosiou et al., 2016) mice, and subsequently bred these double homozygotes (K1$^{fl/fl}$:K2$^{fl/0}$) with Hoxb7-Cre mice (Hoxb7: K1$^{fl/fl}$:K2$^{fl/0}$). Age- and gender-matched littermates without Hoxb7-Cre were used as controls. All experiments were approved by the Vanderbilt University Institutional Animal Use and Care Committee and conducted in AALAC accredited facilities.

### Histology
Mouse kidneys were either fixed in 4% paraformaldehyde overnight before embedding in paraffin wax or they were fixed for 30 min in 4% paraformaldehyde and immersed in sucrose overnight before being embedded in OCT. Kidney sections were stained with Hematoxylin and Eosin, and subjected to light microscopy. Collagen accumulation in kidney sections was evaluated by staining with Picrosirius, ed solution. Proliferation assessment was performed using a Ki-67 antibody (Abcam, ab16667; 1:100). TUNEL staining was performed according to the manufacturer's instructions (Promega Sigma, G7130). Frozen or paraffin wax-embedded kidney sections and primary antibodies were used to stain Six2 (Proteintech, 11562-1-AP; 1:200), E-cadherin (BD Biosciences, 610181; 1:200), phospho-histone H3 (pH3) (Active Motif, 39098; 1:200), anti-β1 integrin (BD Biosciences, 555003; 1:100) and aquaporin 2 (AQP2) (Invitrogen, PA5-77841; 1:100). Alexa Fluor 488 anti-rabbit (ThermoFisher, A21206; 1:200), Alexa Fluor 647 anti-mouse (ThermoFisher, A212236; 1:200), Alexa Fluor 555 anti-rat (ThermoFisher, 78945; 1:200) secondary antibodies and DAPI (Cell Signaling, 4083) were used subsequently. Immunofluorescence images were taken on a Zeiss LSM 980 confocal microscope equipped with an inverted Axio Observer 7 and Airyscan 2 detector. The objective used was a 63×/1.4 numerical aperture (NA) Plan Apochromat oil or 10×/0.50 NA Plan Apochromat (for low-powered scanning and imaging of optically cleared whole-mount tissue).

### Ureteric bud cultures and ectopic budding assay
Wolffian ducts (WDs) were dissected from E10.5 embryos and cultured in DMEM/F12 with 10% FBS as previously described (Chi et al., 2009; Choi et al., 2009; Kuure et al., 2010; Li et al., 2019; Rosines et al., 2010). They were then fixed with 4% PFA for 30 min, washed with PBS and incubated in blocking buffer (2% BSA, 0.2% Tween, 0.2% Triton-X100 and 0.02% sodium azide) for 2-3 h at room temperature. The urogenital systems were stained using an anti-E-cadherin primary antibody (Cell Signaling, 3195; 1:200) followed by a secondary antibody (Alexa Fluor 488 anti-rabbit, ThermoFisher, A21206; 1:200). WD tissue and dissected retroperitoneal

preparations were dehydrated in a methanol gradient and cleared with benzyl alcohol/benzyl benzoate (BABB) solution, as described previously (Bock et al., 2024). Urogenital systems and ureteric buds (UB) were transferred to a glass-bottom dish (Mattek, P35-1.5-14-C) with 100 μl BABB solution for imaging with Zeiss LSM 980 confocal microscopy. Series of z-stack confocal images were loaded into Imaris software (Bitplane) to generate 3D reconstruction and surfaces of WDs. Ureteric trees were analyzed by the filament tracking package in Imaris.

### Generation of CD cells and viability testing
β1 TTAA and β1 Y795A CD cells were generated using a strategy described previously (Mathew et al., 2017). β1-null collecting-duct (CD) cells were transfected with full-length (wild type), TT788/789AA or Y795A mutated human integrin β1 cDNA. The cells were sorted for equal cell surface expression using flow cytometry. CD cells were isolated from the kidneys of 5- to 6-week-old kindlin-1$^{fl/fl}$/kindlin-2$^{fl/fl}$ mice, as previously described (Husted et al., 1988). They were immortalized through SV40 transfection and both kindlin genes were deleted using an adeno-Cre virus. Single clones were generated and tested for successful deletion by western blotting. To test viability cells were stained with Hoechst 33342 (ThermoFisher, 62249) or a Trypan Blue exclusion assay was performed according to standard procedures. Cells were imaged with the ZOE Cell imager (BioRad).

### Flow cytometry
Wild-type or integrin β1 mutant CD cells were incubated with anti-mouse β1 integrin antibody (BD Biosciences, 555003), followed by PE-conjugated secondary antibodies. Expression levels of integrin was detected by flow cytometry and analyzed with floreada.io.

### Tubulogenesis assay
CD cells were grown in collagen/Matrigel gels as previously described (Chen et al., 2004). CD cells ($5 \times 10^3$) were seeded into the gels, which were overlaid with 100 μl of medium and allowed to grow for 7 days. The gels were stained with rhodamine-phalloidin, and the tubules were photographed using a Zeiss confocal microscope (LSM 980).

### Cell adhesion
Cells ($1 \times 10^5$) were plated serum-free on serial dilutions of Matrigel (Corning, 356230) in U-bottom 96-well plates. Adherent cells were fixed, stained with Crystal Violet (Sigma-Aldrich, C6158) and solubilized, and the optical densities of the cell lysates were read at 570 nm ($OD_{570}$).

### Cell migration
Cell migration was assayed as previously described (Chen et al., 2004). Transwells with 8 μm pores were coated with different ECM components, and $1 \times 10^5$ cells were added to the upper well in serum-free medium. Cells that migrated through the filter after 4 h were counted. Eight random fields of each transwell were counted using the particle analyzer function of ImageJ and at least six samples from three independent experiments were analyzed.

### Spreading and focal adhesion visualization
For spreading, 4-well glass slides (Millipore, PEZGS0416) were coated with 1 mg/ml Matrigel (Corning, 356230). $2 \times 10^5$ cells were plated in each well and incubated for 1 h at 37°C 5% $CO_2$. Cells were fixed with 4% paraformaldehyde and stained with rhodamine-conjugated phalloidin (Invitrogen, R415). Cell areas were calculated using ImageJ based on confocal images collected on the Zeiss LSM 980. Quantification of cell area and focal adhesion number was conducted Using ImageJ. Focal adhesions were visualized using immunofluorescence co-staining of rhodamine-conjugated phalloidin (Invitrogen, R415) and phospho-paxillin (Cell Signaling, 2541).

### Cell proliferation
To assess cell proliferation assay, $5 \times 10^3$ cells/well were plated on Matrigel coated 96-well plates in triplicate for each of three independent experiments. After attachment, cells were treated with BrdU (bromodeoxyuridine) for 16 h to label proliferating cells. Cell proliferation was assayed using the BrdU Cell Proliferation Assay Kit (Exalpha, X1327K1) according to the

manufacturer's instructions. After BrdU incorporation, cells were fixed, permeabilized and treated with DNA denaturing solution (HCl) to expose the BrdU. Anti-BrdU antibodies were conjugated with HRP enzyme and incubated with TMB peroxidase substrate and OD was read at 450 nm.

### Cell polarity
Cell polarity assays and polarization index were conducted as previously described (Bock et al., 2021). In brief, $2.5 \times 10^5$ cells were seeded in transwells with 0.4 μm pores (Corning Costar, 3401) and cultured overnight to allow cells to grow to confluence. Cells were fixed in 4% formaldehyde and incubated with ZO-1 antibody (Thermo Fisher, RB231622; 1:200) followed by the AF488-conjugated secondary antibody incubation (ThermoFisher, A21206; 1:200) with the addition of rhodamine-phalloidin (Invitrogen, R415) and DAPI (Cell Signaling, 4083). Chamber slides were mounted and viewed using confocal microscopy and z-stacks were acquired. The ratio of apical to total signal was calculated as the apicalbasal polarization index from at least three independent repeats.

### Replating
For replating, 100 mm cell plate dishes were coated with 1 mg/ml Matrigel (Corning, 356230), Vitronectin (Sigma, SRP3186, at 1.5 μg/ml) or Laminin-511 at 1 μg/ml as previously described (Yazlovitskaya et al., 2019). Plates were blocked with 1% heat-denatured BSA. CD cells were plated on coated dishes for the indicated time points, while unplated cells were used as time 0. After 15, 30 and 45 min, cells were harvested in M-PER reagent (Thermo Fisher, 78501) supplemented with protease and phosphatase inhibitors (protease inhibitor: Sigma, 8340; phosphatase inhibitor cocktail 2: Sigma, 5726; phosphatase inhibitor cocktail 3: Sigma, 0044) and further processed for immunoblotting.

### Growth factor-dependent signaling in vitro and ex vivo
For the cell signaling assay, serum-starved CD cells were stimulated for 24 h after plating with human GDNF recombinant protein (GDNF) 10 ng/ml (Invitrogen, RP-8602) and harvested at the indicated time points. Cells were washed in PBS and lysed using M-PER reagent and subjected to immunoblotting analysis. For the GDNF-dependent ex vivo budding analysis, dissected urogenital systems at E10.5 were cultured in complete medium and treated with 100 ng/ml GDNF. After 48 h the organ cultures were stained with E-cadherin and optically cleared as described above. 3D reconstructions of z-stack confocal images was performed in IMARIS (Bitplane). Three repeats in each mouse genotype were included in statistical analysis.

### Immunoblotting
Lysates were clarified by centrifugation and 30 μg total protein was electrophoresed in gradient 4-20% SDS-PAGE and transferred to nitrocellulose membranes. Membranes were blocked in 3% BSA and incubated with the primary antibody followed by IRDye fluorescent dye secondary antibodies. Immunoreactive bands were imaged with the Li-Cor Biosciences Odyssey system and quantified by densitometry analysis using ImageJ or Image Studio Lite software. Primary antibodies were anti-FAK (Cell Signaling, 3285S; 1:500), anti-pFAK (Cell Signaling, 3283S; 1:500), anti-Pax (Cell Signaling, 2542S; 1:500), anti-pPax (Cell Signaling, 2541S; 1:500), anti-AKT (Cell Signaling, 9272S; 1:500), anti-pAkt (Cell Signaling, 4060S; 1:500), anti-ERK (Cell Signaling, 9102; 1:500), anti-pERK (Cell Signaling, 9101S; 1:500), anti-p38 MAPK (Cell Signaling, 9212S; 1:500) and anti-pp38 MAPK (Cell Signaling, 9211S; 1:500). Secondary antibodies were IRDye 800CW anti-rabbit (Li-COR, 926-32211; 1:1000) and IRDye 680RD anti-rabbit (Li-COR, 926-68071; 1:1000).

### Acknowledgements
We thank Reinhard Faessler for generating and sharing the β1 integrin mutant and kindlin null mice used in this study. We are grateful to the cell imaging shared resource (CISR), to our flow cytometry core at the Nashville VA and to our pathology core (Translational Pathology Shared Resource), and we thank J. Schafer and T. Vaddakan for technical assistance. Microscopy was performed in part using the Vanderbilt Cell Imaging Shared Resource (supported by NIH grants CA68485, DK20593, DK58404, DK59637, EY08126 and 1S10OD034315-01, and the Department of Veteran Affairs). Flow cytometry experiments were performed in

the Nashville VA flow cytometry core (U.S. Department of Veterans Affairs, Tennessee Valley Healthcare System, Nashville, TN). Routine histological processing was performed in our Translational Pathology Shared Resource that is supported by NCI/NIH Cancer Center support grant P30-CA068485.

**Competing interests**
The authors declare no competing or financial interests.

**Author contributions**
Conceptualization: F.B., S.M., R.Z.; Data curation: S.L., A.H., S.M.; Funding acquisition: R.Z.; Investigation: S.L., F.B., O.V., A.H., S.M., R.P., M.M., M.T.; Methodology: F.B., O.V., G.M., X.D.; Resources: R.P., G.M., J.P.A., A.T., T.C., A.P., R.Z.; Supervision: F.B.; Visualization: F.B.; Writing – original draft: F.B., R.Z.; Writing – review & editing: A.P., R.Z.

**Funding**
These studies were supported by the National Institutes of Health (K08 DK134879 to F.B.; DK069921, DK088327 and DK127589 to R.Z.; R01 DK119212 to A.P.), by U.S. Department of Veterans Affairs Merit awards (I01-BX002196 to R.Z. and 1I01BX002025 to A.P.); by a Southern Society for Clinical Investigation Research Scholar Award (to F.B.), by a Kidney Cure PreDoctoral Fellowship (to X.D.) and by a Vanderbilt University Research Scholar Award (to F.B.). A.P. is the recipient of a U.S. Department of Veterans Affairs Senior Research Career Scientist Award (IK6 BX005240). R.Z. is funded by a grant from the W. M. Keck Foundation. Open Access funding provided by Vanderbilt University. Deposited in PMC for immediate release.

**Data and resource availability**
All relevant data and details of resources can be found within the article and its supplementary information.

**Peer review history**
The peer review history is available online at https://journals.biologists.com/dev/lookup/doi/10.1242/dev.205044.reviewer-comments.pdf

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
