## [Peer Review File · Development (Cambridge, England)]

Kindlins regulate integrin- and growth factor-dependent ureteric bud formation

Shensen Li, Fabian Bock, Olga Viquez, Anjana Hassan, Sijo Matthew, Riya Palamuttam, Glenda Mernaugh, Xinyu Dong, Meiling Melzer, Matthew Tantengco, Thomas Carroll, Andrew Terker, Juan Pablo Arroyo, Ambra Pozzi and Roy Zent
DOI: 10.1242/dev.205044

Editor: James M Wells

Review timeline

Original submission:	19 June 2025
Editorial decision:	22 July 2025
First revision received:	10 December 2025
Editorial decision:	16 December 2025
Second revision received:	2 January 2026
Accepted:	6 January 2026

Original submission

First decision letter

MS ID#: dev.205044

MS TITLE: Kindlins regulate integrin- and growth factor-dependent ureteric bud formation

AUTHORS: Shensen Li, Fabian Bock, Olga Viquez, Anjana Hassan, Sijo Matthew, Riya Palamuttam, Glenda Mernaugh, Xinyu Dong, Meiling Melzer, Matthew Tantengco, Juan Pablo Arroyo, Andrew Terker, Thomas Carroll, Ambra Pozzi and Roy Zent

Dear Dr Zent,

I have now received all the referees' reports on the above manuscript, and have reached a decision. The referees' comments are appended below, or you can access them online: please go to:

As you will see, the referees express considerable interest in your work, but have some significant criticisms and recommend a substantial revision of your manuscript before we can consider publication. If you are able to revise the manuscript along the lines suggested, which may involve further experiments, I will be happy to receive a revised version of the manuscript. Your revised paper will be re-reviewed by one or more of the original referees, and acceptance of your manuscript will depend on your addressing satisfactorily the reviewers' major concerns. Please also note that Development will normally permit only one round of major revision. If it would be helpful, you are welcome to contact us to discuss your revision in greater detail. Please send us a point-by-point response indicating your plans for addressing the referees' comments, and we will look over this and provide further guidance.

Please attend to all of the reviewers' comments and ensure that you clearly highlight all changes made in the revised manuscript. Please avoid using 'Tracked changes' in Word files as these are lost in PDF conversion. I should be grateful if you would also provide a point-by-point response detailing how you have dealt with the points raised by the reviewers in the 'Response to Reviewers' box. If you do not agree with any of their criticisms or suggestions please explain clearly why this is so.

Reviewer 1*Advance summary and potential significance to field*

This manuscript by Bock et al. investigates the role of kindlins in kidney ureteric bud (UB) development. Using elegant genetic approaches in mice, the authors dissect the contributions of kindlins to integrin signaling and growth factor responses during UB morphogenesis. They show that disrupting B1 integrin-kindlin interactions via B1 tail mutations results in moderate UB branching defects and collecting duct abnormalities, while UB-specific deletion of kindlin-1/2 completely abrogates kidney development. Complementary in vitro and ex vivo experiments support a dual role of kindlins in both integrin- and growth factor-mediated pathways. The study addresses an important question in developmental biology, building logically on their prior work on talins and integrins. This is an excellent manuscript: the data are convincing and of interest to readers in the fields of kidney development, integrin signaling, and morphogenesis.

Comments on areas that could be improved and/or clarified:

1. The authors examine HoxB7-cre; Itgb1fl/YA and HoxB7-cre; Itgb1fl/TTAA mice, but the genotype of the control animals is not specified. Please indicate the specific control genotype used for each experiment. Additionally, do HoxB7-cre; Itgb1fl/+ mice have any phenotype? This information is important for interpreting the severity of the observed defects.
2. While the manuscript states that B1 integrin surface expression in mutant CD cells is equivalent to controls (data not shown), it would strengthen the conclusions to include the immunostaining for B1 integrin in kidney sections and/or CD cells. This would demonstrate that B1 expression and localization are maintained in the mutants despite the point mutations.
3. The statement, "Of note, there was also no dilatation of the tubules indicating that the CD tubular pathology only occurred after birth," may overstate the conclusion.
4. Please specify how many embryos or mice were analyzed per genotype for key in vivo experiments.
5. In Figures 6B and 6E, kindlin1/2-null CD cells appear rounded and poorly adherent, raising the question of whether these cells remain viable. Have the authors performed viability assays (e.g., live/dead staining, trypan blue exclusion) to confirm that the observed defects are not secondary to cell death?
6. Given that both kindlin-1 and -2 are deleted, the relative contributions of each remains unknown. The authors could discuss the relative expression of these in the UB and whether single knockouts show any phenotype or whether one isoform is dominant.
7. The authors conclude that kindlins mediate GDNF-dependent signaling independently of B1 integrins based largely on the finding that kindlin-null cells do not respond to GDNF even on plastic, while B1 mutants do. However, the mechanistic basis of this conclusion is unclear and the impaired GDNF signaling in kindlin-null cells could be secondary to severe cellular or adhesion/cytoskeletal defects. The authors should discuss this.

Reviewer 2*Advance summary and potential significance to field*

In this manuscript, the authors investigate the role of Kindlins in kidney development using genomics-based experiments in mice. They demonstrate that double knockout of Kindlin-1 and Kindlin-2 in the Wolffian duct results in a complete failure of ureteric bud formation. Mechanistically, they propose that the Wolffian duct lacking Kindlin1/2 may lose its responsiveness to GDNF signaling. Furthermore, they show that introducing a point mutation in the Kindlin-binding site of integrin B1 suppresses ureteric bud branching and leads to cell death in the collecting duct epithelium in the postnatal kidney. This study builds upon the authors' previous work and provides valuable insight into the critical role of integrins in kidney development. However, to further solidify the novelty of the findings, the following points warrant additional experimental validation:

Comments for the author

Point 1:

Although the kidney developmental phenotype in the Kindlin1/2 double mutant was severe, the phenotype observed in the integrin-β1 point mutation was relatively mild. Further investigation is needed to explain this discrepancy, as understanding this difference is central to the novelty of the current study. Specifically, the authors should explore whether integrin-β1 is indeed the most critical β integrin for ureteric bud formation, or if other β integrin family members might play redundant or compensatory roles. This could be addressed by introducing similar point mutations into other integrin β family genes to test for functional redundancy.

Point 2:

In their experiments using collecting duct cells, the authors examine phosphorylation of various proteins that may be downstream targets of integrin signaling. However, they use Matrigel, a complex mixture of extracellular matrix (ECM) components (mainly collagen I), which makes it difficult to determine which integrins are actually being activated. Indeed, as shown in the authors' previous study (Development, 2009), integrin signaling can still be activated in integrin-β1 knockout collecting duct cells when cultured on vitronectin. Therefore, instead of using a mixed ECM like Matrigel, the authors should systematically select individual ECM components that serve as potential integrin ligands and test them separately. This approach would help clarify which β integrin family members are most critical for ureteric bud formation.

First revisionAuthor response to reviewers' comments

Dear Professor James M Wells,

Thank you for the positive reviews for our paper titled "*Kindlins regulate integrin- and growth factor-dependent ureteric bud formation*". (manuscript number: MS ID#: dev.205044). We appreciate the helpful and constructive comments from you and the reviewers, and we have addressed all the reviewer's comments. We believe the manuscript is significantly improved and we thank you again for the opportunity to revise and resubmit our paper. Below is our point-by-point response to the reviewers comments.

Reviewer 1

Overall the comments from this reviewer were positive. We have addressed the comments that could be improved or clarified below:

1. The authors examine HoxB7-cre; Itgb1fl/YA and HoxB7-cre; Itgb1fl/TTAA mice, but the genotype of the control animals is not specified. Please indicate the specific control genotype used for each experiment. Additionally, do HoxB7-cre; Itgb1fl/+ mice have any phenotype? This information is important for interpreting the severity of the observed defects.

As controls we utilized Itgb1^{fl/wt}/Y/A or Itgb1^{fl/wt}/TT/AA mice. HoxB7-cre; Itgb1^{fl/wt}, Itgb1^{fl/wt}/Y/A and Itgb1^{fl/wt}/TT/AA mice do not have any obvious phenotype. This has now been added to the results section on page 5.

2. While the manuscript states that β1 integrin surface expression in mutant CD cells is equivalent to controls (data not shown), it would strengthen the conclusions to include the immunostaining for β1 integrin in kidney sections and/or CD cells. This would demonstrate that β1 expression and localization are maintained in the mutants despite the point mutations.

We thank you for this comment. We now show integrin expression in vivo and in vitro in supplemental figure 1. The results are described on page 5 and page 6.

3. The statement, "Of note, there was also no dilatation of the tubules indicating that the CD tubular pathology only occurred after birth," may overstate the conclusion.

We have deleted this sentence in the revised version.

4. Please specify how many embryos or mice were analyzed per genotype for key in vivo experiments.

We have indicated the mouse numbers in the figure legends.

5. In Figures 6B and 6E, kindlin1/2-null CD cells appear rounded and poorly adherent, raising the question of whether these cells remain viable. Have the authors performed viability assays (e.g., live/dead staining, trypan blue exclusion) to confirm that the observed defects are not secondary to cell death?

We thank the reviewer for this comment. To address the aspect of cell viability of the K1/K2^{-/-} CD cells we have performed Hoechst staining and a trypan blue exclusion assay on the mutant and WT cells. This is represented in supplemental figure 3 and described on page 8.

6. Given that both kindlin-1 and -2 are deleted, the relative contributions of each remains unknown. The authors could discuss the relative expression of these in the UB and whether single knockouts show any phenotype or whether one isoform is dominant.

We thank the reviewer for this comment and we have stated the role of the different isoforms is currently unknown in the rewritten discussion (page 13). We are currently putting together a manuscript on the specific roles of the two kindlins where we demonstrate how they mediate their different functions in the developing ureteric bud. This work will be ready for publication soon.

7. The authors conclude that kindlins mediate GDNF-dependent signaling independently of B1 integrins based largely on the finding that kindlin-null cells do not respond to GDNF even on plastic, while B1 mutants do. However, the mechanistic basis of this conclusion is unclear and the impaired GDNF signaling in kindlin-null cells could be secondary to severe cellular or adhesion/cytoskeletal defects. The authors should discuss this.

We thank you for this comment and agree with it. We have now included this in the rewritten discussion (page 11). Also please see comment to point 1 of reviewer 2.

Reviewer 2: SUMMARY OF THE ADVANCE MADE IN THIS PAPER AND ITS POTENTIAL SIGNIFICANCE TO THE FIELD

In this manuscript, the authors investigate the role of Kindlins in kidney development using genomics-based experiments in mice. They demonstrate that double knockout of Kindlin-1 and Kindlin-2 in the Wolffian duct results in a complete failure of ureteric bud formation. Mechanistically, they propose that the Wolffian duct lacking Kindlin1/2 may lose its responsiveness to GDNF signaling. Furthermore, they show that introducing a point mutation in the Kindlin-binding site of integrin B1 suppresses ureteric bud branching and leads to cell death in the collecting duct epithelium in the postnatal kidney. This study builds upon the authors' previous work and provides valuable insight into the critical role of integrins in kidney development. However, to further solidify the novelty of the findings, the following points warrant additional experimental validation:

SUGGESTIONS TO AUTHORS

Point 1:

Although the kidney developmental phenotype in the Kindlin1/2 double mutant was severe, the phenotype observed in the integrin-B1 point mutation was relatively mild. Further investigation is needed to explain this discrepancy, as understanding this difference is central to the novelty of the current study. Specifically, the authors should explore whether integrin-B1 is indeed the most critical B integrin for ureteric bud formation, or if other B integrin family members might play redundant or compensatory roles. This could be addressed by introducing similar point mutations into other integrin B family genes to test for functional redundancy.

We thank the reviewer for this comment. We addressed this possibility by making compound beta1/alpha v mutants in the ureteric bud. Double homozygotes mice do not form kidneys and are totally anephric. Alpha v null mice have normal ureteric bud development and beta1 homozygote/alphav heterozygotes have a worse phenotype than beta homozygote mice. These

data have been included in supplemental figure 6 and discussed on page 11 in the discussion of the resubmitted manuscript.

Point 2:

In their experiments using collecting duct cells, the authors examine phosphorylation of various proteins that may be downstream targets of integrin signaling. However, they use Matrigel, a complex mixture of extracellular matrix (ECM) components (mainly collagen I), which makes it difficult to determine which integrins are actually being activated. Indeed, as shown in the authors' previous study (Development, 2009), integrin signaling can still be activated in integrin- β 1 knockout collecting duct cells when cultured on vitronectin. Therefore, instead of using a mixed ECM like Matrigel, the authors should systematically select individual ECM components that serve as potential integrin ligands and test them separately. This approach would help clarify which β integrin family members are most critical for ureteric bud formation.

We thank the reviewer for this comment, and we have performed replating assays on collagen 1, laminin 511 (beta 1 specific ligands) and vitronectin (alpha v specific ligand). Using these matrices we still do not see major differences between WT and the beta1 mutants. We also used these same substrates for the K1/K2 cells and noted that these cells also behaved the same as when we used Matrigel. Together these data suggest that point mutations in the beta1 integrin CT minimally affect adhesion-dependent signaling. By contrast deleting K1/K2 diminishes both beta1 integrin and alpha v integrin adhesion-dependent signaling. These data are presented in supplemental figures 2 and 4.

Second decision letter

MS ID#: dev.205044R1

MS TITLE: Kindlins regulate integrin- and growth factor-dependent ureteric bud formation

AUTHORS: Shensen Li, Fabian Bock, Olga Viquez, Anjana Hassan, Sijo Matthew, Riya Palamuttam, Glenda Mernaugh, Xinyu Dong, Meiling Melzer, Matthew Tantengco, Juan Pablo Arroyo, Andrew Terker, Thomas Carroll, Ambra Pozzi and Roy Zent

Dear Dr Zent,

I have now received all the referees reports on the above manuscript, and have reached a decision. The referees' comments are appended below.

The overall evaluation is positive and we would like to publish a revised manuscript in Development, provided that the referees' minor comments can be satisfactorily addressed. The two remaining reviewer comments are important to address in your revised manuscript and please detail them in your point-by-point response. If you do not agree with any of their criticisms or suggestions explain clearly why this is so. If it would be helpful, you are welcome to contact us to discuss your revision in greater detail. Please send us a point-by-point response indicating your plans for addressing the referees' comments, and we will look over this and provide further guidance.

Reviewer 2

Advance summary and potential significance to field

The authors have almost appropriately addressed my previous comments. In their newly generated figures and descriptions, I only found 2 minor points to be corrected.

1. Regarding to discussion about Supplementary Figure 6, as data in Supplementary Figure 6 are results, those results should be presented in the result section but not in the discussion section.
 2. Regarding to Supplementary Figure 6, please provide a table or a graph showing the quantified data about the number of normal kidneys / hypoplastic kidneys / kidney agenesis developed in each condition including the control, B1 +/fl:av +/fl, B1 fl/fl:av +/fl, and B1 fl/fl:av fl/fl.
-

Second revision

Author response to reviewers' comments

Dear Professor James M Wells,

Thank you for the positive reviews for our paper titled “*Kindlins regulate integrin- and growth factor-dependent ureteric bud formation*”. (manuscript number: MS ID#: dev.205044). We appreciate the helpful and constructive comments from you and the reviewers, and we have addressed all the reviewer’s comments. We believe the manuscript is significantly improved and we thank you again for the opportunity to revise and resubmit our paper. Below is our point-by-point response to the reviewers’ comments.

Reviewer 2

1. Regarding to discussion about Supplementary Figure 6, as data in Supplementary Figure 6 are results, those results should be presented in the result section but not in the discussion section.

2. Regarding to Supplementary Figure 6, please provide a table or a graph showing the quantified data about the number of normal kidneys / hypoplastic kidneys / kidney agenesis developed in each condition including the control, B1 +/fl:av +/fl, B1 fl/fl:av +/fl, and B1 fl/fl:av fl/fl.

Response

We agree with the reviewer on this point, and we have split the results revolving around the data from supplementary figure 6 into a results section and a discussion. These sections are found on pages 9 and 12 respectively.

Third decision letter

MS ID#: dev.205044R2

MS TITLE: Kindlins regulate integrin- and growth factor-dependent ureteric bud formation

AUTHORS: Shensen Li, Fabian Bock, Olga Viquez, Anjana Hassan, Sijo Matthew, Riya Palamuttam, Glenda Mernaugh, Xinyu Dong, Meiling Melzer, Matthew Tantengco, Juan Pablo Arroyo, Andrew Terker, Thomas Carroll, Ambra Pozzi and Roy Zent

Dear Dr Zent,

I am happy to tell you that your manuscript has been accepted for publication in Development, pending our standard publication integrity checks.